

# A fast vectorized sorting implementation based on the ARM scalable vector extension (SVE)

Bérenger Bramas

CAMUS, Inria Nancy - Grand Est, Nancy, France
ICPS Team, ICube, Illkirch-Graffenstaden, France

## ABSTRACT

The way developers implement their algorithms and how these implementations behave on modern CPUs are governed by the design and organization of these. The vectorization units (SIMD) are among the few CPUs' parts that can and must be explicitly controlled. In the HPC community, the x86 CPUs and their vectorization instruction sets were de-facto the standard for decades. Each new release of an instruction set was usually a doubling of the vector length coupled with new operations. Each generation was pushing for adapting and improving previous implementations. The release of the ARM scalable vector extension (SVE) changed things radically for several reasons. First, we expect ARM processors to equip many supercomputers in the next years. Second, SVE's interface is different in several aspects from the x86 extensions as it provides different instructions, uses a predicate to control most operations, and has a vector size that is only known at execution time. Therefore, using SVE opens new challenges on how to adapt algorithms including the ones that are already well-optimized on x86. In this paper, we port a hybrid sort based on the well-known Quicksort and Bitonic-sort algorithms. We use a Bitonic sort to process small partitions/arrays and a vectorized partitioning implementation to divide the partitions. We explain how we use the predicates and how we manage the non-static vector size. We also explain how we efficiently implement the sorting kernels. Our approach only needs an array of $O(\log N)$ for the recursive calls in the partitioning phase, both in the sequential and in the parallel case. We test the performance of our approach on a modern ARMv8.2 (A64FX) CPU and assess the different layers of our implementation by sorting/partitioning integers, double floating-point numbers, and key/value pairs of integers. Our results show that our approach is faster than the GNU C++ sort algorithm by a speedup factor of 4 on average.

## INTRODUCTION

Sorting is a fundamental problem in computer science and a critical building block for many types of applicationssuch as, but not limited to, database servers (*Graefe, 2006*), image rendering engines (*Bishop et al., 1998*), mining of time series (*Raoofy et al., 2020*) or JPEG steganography with particle swarm (*Snasel et al., 2020*). This has pushed the research community to spend many efforts to provide efficient sorting libraries on new architectures.

Corresponding author
Bérenger Bramas,
berenger.bramas@inria.fr

This research of performance is coupled to the changes in CPUs' designs and organization, which includes the vectorization capability.

The performance of CPUs has improved for several decades by increasing the clock frequency. However, this approach has reached a steady-state due to power dissipation and heat effects. To go beyond this limitation, the manufacturers have used parallelization at multiple levels: by embedding multi-cores in a CPU, by allowing pipelining and out-of-order execution at the instruction-level, by putting multiple computational units in each core, and by supporting vectorization. In this context, vectorization consists in the capability of a CPU core of applying a single instruction on multiple data, a concept called SIMD by Flynn's taxonomy (*Flynn, 1972*). The consequence of this hardware design is that it is mandatory to vectorize a code to achieve high-performance. On the contrary, the throughput can be reduced by at least a factor equivalent to the length of a vector compared to the theoretical peak of the hardware. For instance, vector sizes for single precision values are 8 in widespread CPUs (AVX2) and 16 (AVX-512) on many computing nodes.

We can convert many classes of algorithms and computational kernels from a scalar code into a vectorized equivalent without difficulties. Besides, it can be done with auto-vectorization for some of them. However, some algorithms are challenging to adapt because of their memory/data access patterns. Data-processing algorithms (like sorting) are of this kind and require a significant programming effort to be vectorized efficiently. Also, the possibility of creating a fully vectorized implementation, with no scalar sections and with few data transformations, is only possible and efficient if the instruction set extension (IS) provides the needed operations. This is why new ISs together with their new operations make it possible to invent approaches that were not feasible previously, at the cost of reprogramming.

Vectorizing a code can be described as solving a puzzle, where the board is the target algorithm and the pieces are the size of the vector and the instructions. However, the paradigm changes with SVE (*Stephens et al., 2017*; *ARM, 2020b*; *ARM, 2020a*) because the size of the vector is unknown at compile time. This can have a significant impact on the transformation from scalar to vectorial. As an example, consider that a developer wants to work on a fixed number of values, which could be linked to the problem to solve, *e.g.*, a $16 \times 16$ matrix-matrix product, or based on other references, *e.g.*, the size of the L1 cache. When the size of the vector is known at development time, a block of data can be mapped to the corresponding number of vectors and working on the vectors can be done with static/known number of operations. With a variable size, it is required to either implement different kernels for each of the possible sizes (like if they were different ISs) or by finding a generic way to vectorize the kernel, which could be a tedious task. We could expect SVE to be less upgraded than *x86* ISs because there will be no need to release a new IS even when new CPU generations will support larger vectors.

In the current paper, we focus on the adaptation of a sorting strategy and its efficient implementation for the ARM CPUs with SVE. Our implementation is generic and works for any size equal to a power of two. The contributions of this study are:

- Describe how we port our AVX-SORT algorithm (*Bramas, 2017*) to SVE;

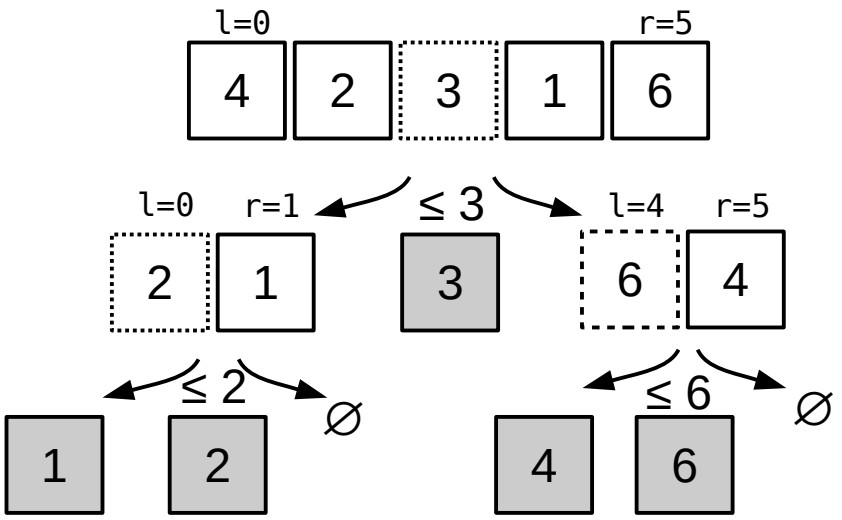

**Figure 1 Quicksort algorithm example.** Quicksort example to sort [3,1,2,0,5] to [0,1,2,3,5]. The pivot is equal to the value in the middle: the first pivot is 2, then at second recursion level it is 1 and 5.

[1] The current study provides a translation of our AVX-SORT into SVE but also a completely new approach which works when the vector size is unknown at compile time.

[2] The functions described in the current study are available at https://gitlab.inria.fr/bramas/sve-sort. This repository includes a clean header-only library and a test file that generates the performance study of the current manuscript. The code is under MIT license.

- Define a new Bitonic-sort variant using SVE and how runtime vector size impact the implementation[1];
- Implement an efficient Quicksort variant using OpenMP (*Board, 2013*) tasks.

All in all, we show how we can obtain a fast and vectorized in-place sorting implementation[2].

The paper is organized as follows: We first give background information related to vectorization and sorting in 'Background'. Then, in 'Sorting with SVE', we describe our strategies for sorting small arrays, for partitioning and for our parallel sort. Finally, the performance study is detailed in 'Performance study'.

## BACKGROUND

### Sorting algorithms
#### *Quicksort (QS) overview*

QS (*Hoare, 1962*) is a sorting algorithm that followed a *divide-and-conquer* strategy: the input array is recursively partitioned until the partitions hold a single element. The partitioning algorithm moves the values lower than a *pivot* at the beginning of the array, and greater values at the end, with a linear complexity. The worst-case complexity of QS is $O(n^2)$, but in practice it has an average complexity of $O(n \log n)$. The complexity is tied to the pivot, and it must be close to the median to ensure a low complexity. However, it is a very popular sorting algorithm thanks to its simplicity in terms of implementation, and its speed in practice. An example of a QS execution is provided in Fig. 1.

To parallelize the QS and other *divide-and-conquer* approaches, it is common to create a task for each recursive call followed by a wait statement. For instance, a thread partitions the array in two, and then creates two tasks (one for each of the partition). To ensure

coherency, the thread waits for the completion of the tasks before continuing. We refer to this parallel strategy as the *QS-par*.

### GNU std::sort implementation (STL)

The standard requires a worst-case complexity of O($n$ log $n$) (*ISO, 2014*) (it was an average complexity until year 2003 (*ISO, 2003*) that a pure QS implementation cannot guarantee. Consequently, the QS algorithm cannot be used alone as a standard *C++* sort. As a result, the current STL implementation relies on 3 different algorithms[3]. This i3-part hybrid sorting algorithm is composed of an Introsort (*Musser, 1997*) to a maximum depth of $2 \times \log^2 n$ to obtain small partitions. These partitions are then sorted using an insertion sort, which is a 2-part hybrid composed of Quicksort and Heapsort.

### Bitonic sorting network

In computer science, a sorting network is an abstract that describes how the values to sort are compared and exchanged. A network is defined for a given number of values. It is possible to represent graphically a sorting network where horizontal lines represent the input values, and vertical connection between those lines represent *compare and exchange* units. The literature provides various examples of sorting networks, and our approach relies on the Bitonic sort (*Batcher, 1968*). This network is straightforward to implement and its algorithm complexity for any input is of O($n \log(n)^2$). This algorithm demonstrated good performances on parallel computers (*Nassimi & Sahni, 1979*) and GPUs (*Owens et al., 2008*). We provide a Bitonic sorting network to sort 16 values in Fig. 2A. The execution of the example goes from left to right as a timeline. The values are moved from left to right, and when they cross an exchange unit they are potentially transferred along the vertical bar. Figure 2B provide a real example where we print the values along the horizontal lines when sorting eight values. We use the terms *symmetric* and *stair* exchanges to refer to the red and orange stages, respectively. A *symmetric* stage is always followed by *stair* stages from half size to size two. The Bitonic sort does not maintain the original order of the values and thus is not stable.

We can implement a sorting network by hard-coding the connections between the lines only if we know the size of the input array. In this case, we simply translate the picture into an algorithm. However, for a dynamic array size, the implementation has to be flexible by relying on formulas that define when the lines cross (*Grama et al., 2003*).

## Vectorization

The word vectorization defines a CPU capability of applying a single operation/instruction to a vector of values, instead of a single/scalar value (*Kogge, 1981*). Thanks to this feature, the peak performance of single cores had continued to increase despite the stagnation of the clock frequency since the mid-2000s. In the meanwhile, the length of the SIMD registers (*i.e.,* the size of the vectors) has continuously increased, which increases the performance of the chips accordingly. In the current study, the term *vector* has no relation to an expandable vector data structure, such as *std::vector*, but refers to the data type managed by the CPU in this sense. The size of the vectors is variable and depends on both the instruction set and the type of vector's elements, and corresponds to the size of the registers in the chip.

[3] See the libstdc++ documentation on the sorting algorithm available at https://gcc.gnu.org/onlinedocs/libstdc++/libstdc++-html-USERS-4.4/a01347.html#l05207.

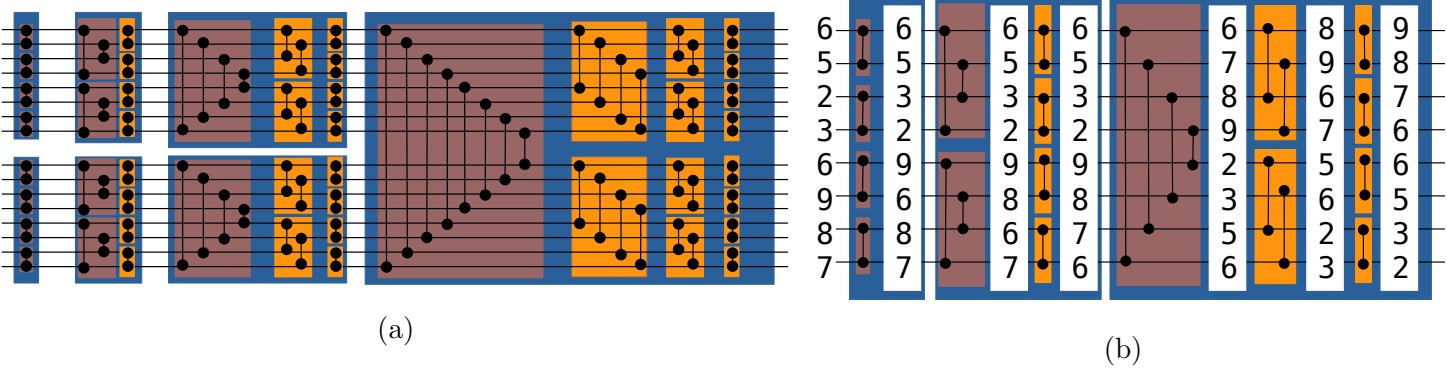

(a)

(b)

**Figure 2** **(A-B) Bitonic sorting network examples.** Bitonic sorting network examples. In red boxes, the exchanges are done from extremities to the center and we refer to it as the symmetric stage, whereas in orange boxes, the exchanges are done with a linear progression and we refer to it as the stair stage.

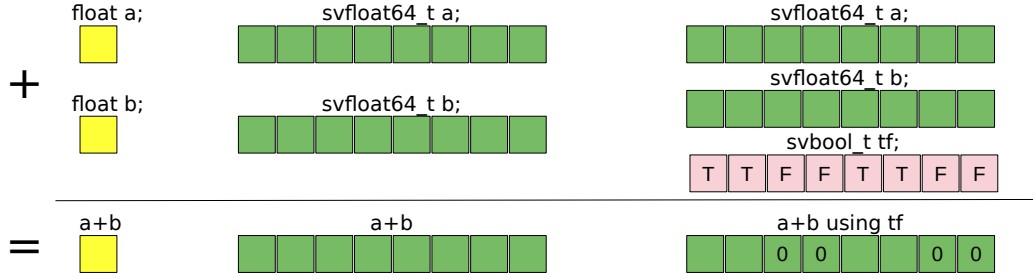

**Figure 3** **Summation example of single precision floating-point values.** Summation example of single precision floating-point values using: (yellow) scalar standard C++ code, (red) SSE SIMD-vector of four values, (green) AVX SIMD-vector of eight values.

The SIMD instructions can be called in the assembly language or using *intrinsic* functions, which are small functions that are intended to be replaced with a single assembly instruction by the compiler. There is usually a one-to-one mapping between intrinsics and assembly instructions, but this is not always true, as some intrinsics are converted into several instructions. Moreover, the compiler is free to use different instructions as long as they give the same results.

The SVE is a feature for ARMv8 processors. The size of the vector is not fixed at compile time (the specification limits the size to 2048 bits and ensures that it is a multiple of 128 bits) such that a binary that includes SVE instructions can be executed on ARMv8 that support SVE no matter the size of their registers. Figure 3 illustrates the difference between a scalar summation and a vector summation with a vector size of 256 bits.

SVE provides most classic operations that also exist in *x86* vectorization extensions, such as loading a contiguous block of values from the main memory and transforming it into a SIMD-vector (load), filling a SIMD-vector with a value (set), move back a SIMD-vector into memory (store) and basic arithmetic operations. SVE also provides advanced operations

like gather, scatter, indexed accesses, permutations, comparisons, and conversions. It is also possible to get the maximum or the minimum of a vector or element-wise between two vectors.

Another significant difference with other ISs, is the use of predicate vectors *i.e.* the use of Boolean vectors (*svbool_t*) that allow controlling more finely the instructions by selecting the affected elements, for example. Also, while in AVX-512 the value returned by a test/comparison (*vpcmpd/vcmppd*) is a mask (integer), in SVE, the result is *svbool_t*.

A minor difference, but which impacts our implementation, is that SVE does not support a *store-some* as it exists in AVX-512 (*vpcompressps/vcompresspd*), where some values of a vector can be stored contiguously in memory. With SVE it is needed to first compact the values of a vector to put the values to be saved at the beginning of the vector, and then perform a store, or to use a scatter. However, both approaches need extra Boolean or indices vectors and additional instructions.

## Related work on vectorized sorting algorithms

The literature on sorting and vectorized sorting implementations is very large. Therefore, we only cite some studies we consider most related to our work.

*Sanders & Winkel (2004)* provide a sorting technique that tries to remove branches and improves the prediction of a scalar sort. The results show that the method provides a speedup by a factor of 2 against the STL (the implementation of the STL was different). This study illustrates the early strategy to adapt sorting algorithms to a given hardware, and also shows the need for low-level optimizations, due to the limited instructions available.

Later, *Inoue et al. (2007)* propose a parallel sorting on top of combosort vectorized with the VMX instruction set of IBM architecture. Unaligned memory access is avoided, and the L2 cache is efficiently managed by using an out-of-core/blocking scheme. The authors show a speedup by a factor of 3 against the GNU *C++* STL.

In a different study (*Furtak, Amaral & Niewiadomski, 2007*), Furtak et al. use a sorting-network for small-sized arrays, similar to our own approach. However, instead of dividing the main array into sorted partitions (partitions of increasing contents), and applying a small efficient sort on each of those partitions, the authors perform the opposite. They apply multiple small sorts on sub-parts of the array, and then they finish with a complicated merge scheme using extra memory to sort globally all the sub-parts. A very similar approach was later proposed by *Chhugani et al. (2008)*. More recently, *Gueron & Krasnov (2016)* provided a new approach for AVX2. The authors use a Quicksort variant with a vectorized partitioning function, and an insertion sort once the partitions are small enough (as the STL does). The partition method relies on look-up tables, with a mapping between the comparison's result of an SIMD-vector against the pivot, and the move/permutation that must be applied to the vector. The authors show a speedup by a factor of 4 against the STL, but their approach is not always faster than the Intel IPP library. The proposed method is not suitable for AVX-512 because the lookup tables will occupy too much memory. This issue and the use of extra memory, can be solved with the new instructions of the AVX-512. As a side remark, the authors do not compare their proposal to the standard *C++ partition* function. It is the only part of their algorithm that is vectorized.

In our previous work (*Bramas, 2017*), we have proposed the first hybrid QS/Bitonic algorithm implemented with AVX-512. We have described how we can vectorize the partitioning algorithm and create a branch-free/vectorized Bitonic sorting kernel. To do so, we put the values of the input array into SIMD vectors. Then, we sort each vector individually, and finally we exchange values between vectors either during the *symmetric* or *stair* stage. Our method was eight times faster to sort small arrays and 1.7 times faster to sort large arrays compared to the Intel IPP library. However, our method was sequential and could not simply be converted to SVE when we consider that the vector size is unknown at compile time. In this study, we refer to this approach as the *AVX-512-QS*.

*Hou, Wang & Feng (2018)* designed a framework for the automatic vectorization of parallel sort on *x86*-based processors. Using a DSL, their tool generates a SIMD sorting network based on a formula. Their approach shows a significant speedup against STL, and especially they show a speedup of 6.7 in parallel against the sort from Intel TBB on Intel Knights Corner MIC. The method is of great interest as it avoids programming by hand the core of the sorting kernel. Any modification, such as the use of a new IS, requires upgrading the framework. To the best of our knowledge, they do not support SVE yet.

*Yin et al. (2019)* described an efficient parallel sort on AVX-512-based multi-core and many-core architectures. Their approach achieves to sort 1.1 billion floats per second on an Intel KNL (AVX-512). Their parallel algorithm is similar to the one we use in the current study because they first sort sub-parts of the input array and then merge them by pairs until there is only one result. However, their parallel merging is out-of-place and requires doubling the needed memory, which is not the case for us. Besides, their Bitonic sorting kernel differs from ours, because we follow the Bitonic algorithm without the need for matrix transposition inside the registers.

*Watkins & Green (2018)* provide an alternative approach to sort based on the merging of multiple vectors. Their method is two times faster than the Intel IPP library and 5 times faster than the C-lib *qsort*. They can sort 500 million keys per second on an Intel KNL (AVX-512) but they also need to have an external array when merging, which we avoid in our approach.

## Related work on vectorized with SVE

Developing optimized kernels with SVE is a recent research topic. We refer to studies that helped better understand this architecture, even if they did not focus on sorting.

*Meyer et al. (2018)* studied the assembly code generated when implementing lattice quantum chromodynamics (LQCD) kernels. They evaluate if the compiler was capable of generating vectorized assembly from a scalar C code, which was the case. LQCD has also been studied by *Alappat et al. (2021)* in addition to sparse matrix vector product (SpMV). The authors studied various effects and properties of the A64FX, and demonstrated that for some kernels it competes with a V100 GPU.

*Kodama et al. (2017)* tried evaluating the impact on performance when changing the vector size, while using the same hardware. Their objective was oriented to SVE since SVE kernels are vector size independent. At the time of the study, no hardware was supported SVE, hence the authors used an emulator. Additionally, they speculated on how the vector

size could be changed, which does not respect the current SVE technology. Nevertheless, they concluded that using a larger vector than the hardware registers could be beneficial, by reducing the number of instructions, but could also be negative by the need of more registers.

*Aoki & Murao (0000)* implemented the H.265 video codec using SVE. Their implementation reduces the number of instructions by half, but no performance results were given.

*Wan, Gu & Su (2021)* implemented level-2 basic linear algebra routines (BLAS). They evaluated their implementation using the Arm emulator ARMIE, which predicted a 17x speedup against Neon, the previous ARM vector ISA.

*Domke (2021)* evaluated the performance differences for various benchmarks and five different compilers on A64FX. The author advised Fujitsu for Fortran codes, GNU for integer-intensive apps, and any clang-based compilers for C/C++, but concluded that there was not a single perfect compiler and that it is advised to test for each application.

## SORTING WITH SVE

### Overview

Our SVE-QS shares similarities with the AVX-512-QS as it is composed of two key steps. First, we partition the data recursively using the *sve_partition* function described in 'Partitioning with SVE', as in the classical QS. Second, once the partitions are smaller than a given threshold, we sort them with the *sve_bitonic_sort_wrapper* function from 'Bitonic-based sort on SVE vectors'. To sort in parallel, we rely on the classical parallelization scheme for the divide-and-conquer algorithm, but propose several optimizations. This allows an easy parallelization method, which can be fully implemented using OpenMP.

### Bitonic-based sort on SVE vectors

In this section, we detail our Bitonic-sort to sort small arrays that have less than 16 times *VEC_SIZE* elements, where *VEC_SIZE* is the size of a SIMD vector. We used this function in our QS implementation to sort partitions that are small enough.

#### *Sorting one vector*

We sort a single vector by applying the same operations as the ones shown in Fig. 2A. We perform the compare and exchange following the indexes shown in the Bitonic sorting network figure. Thanks to the vectorization, we are able to work on an entire vector without the need of iterating on the values individually. However, we cannot hard-code the indices of the elements that should be compared and exchanged, because we do not know the size of the vector. Therefore, we use a loop-based scheme where we efficiently generate permutation and Boolean vectors to perform the correct comparisons. We use the same pattern for both the *symmetric* and the *stair* stages.

In the *symmetric* stage, the values are first compared by contiguous pairs, e.g. each value at an even index $i$ is compared with the value at $i + 1$ and each value at en odd index $j$ is compared with the value at $j - 1$. Additionally, we see in Fig. 2A that the width of comparison doubles at each iteration and that the comparisons are from the sides to

the center. In our approach, we use three vectors. First, a Boolean vector that shows the direction of the comparisons, e.i. for each index it tells if it has to be compared with a value at a greater index (and will take the minimum of both) or with a value at a lower index (and will take the maximum of both). Second, we need a shift coefficient vector which gives the step of the comparisons, *i.e.* it tells for each index the relative position of the index to be compared with. Finally, we need an index vector that contains increasing values from 0 to N-1 (for any index $i$, $vec[i] = i$) and that we use to sum with the shift coefficient vector to get a permutation vector. The permutation vector tells for any index which other index it should be compared against.

We give the pseudo-code of our vectorized implementation in Algorithm 1 where the corresponding SVE instructions and possible vector values are written in comments. In the beginning, the Boolean vector must contain repeated *false* and *true* values because the values are compared by contiguous pairs. To build it, we use the *svzip1* instruction which interleaves elements from low halves of two inputs, and pass one vector of *true* and a vector of *false* as parameters (line 7). Then, at each iteration, the number of *false* should double and be followed by the same number of *true*. To do so, we use again the *svzip1* instruction, but we pass the Boolean vector as parameters (line 7). The vector of increasing indexes is built with a single SVE instruction (line 5). The shift coefficients vector is built by interleaving 1 and $-1$ (line 9). The permutation index is generated by summing the two vectors (line 12) and uses to permute the input (line 14). So, at each iteration, we use the updated Boolean vector to decide if we add or subtract two times the iteration index (line 22). Also, this algorithm is never used as presented here because each of its iterations must be followed by a *stair* stage.

---

**Algorithm 1:** SVE Bitonic sort for one vector, symmetric stage.

**Input**: vec: a SVE vector to sort.
**Output**: vec: the vector sorted.

```
 1  function sve_bitonic_sort_1v_symmetric(vec)
 2      //Number of values in a vector
 3      N = get_hardware_size()
 4      //svindex - [O, 1, …., N-1]  vecIndexes = (i ∈ [0, N-1] → i)
 5      //svzip1 - [F, T, F, T, …, F, T]
 6      falseTrueVecOut = (i ∈ [0, N-1] → i is odd ? False: True)
 7      //svneg/svdup - [1, -1, 1, -1, …, 1, -1]
 8      vecIndexesPermOut = (i ∈ [0, N-1] → falseTrueVecOut[i] ? -1: 1)
 9      for stepOut from 1 to N-1, doubling stepOut at each step do
10          //svadd - [1, 0, 3, 2, …, N-1, N-2]
11          premuteIndexes = (i ∈ [0, N-1] → vecIndexes[i] + vecIndexesPermOut[i])
12          //svtbl - [vec[1], vec[0], vec[3], vec[2], …, vec[N-1], vec[N-2]]
13          vecPermuted = (i ∈ [0, N-1] → vec[premuteIndexes[i]])
14          //svsel/svmin/svmax - [..., Min(vec[i], vec[i+1]), Max(vec[i], vec[i+1]), …]
15          vec = (i ∈ [0, N-1] → falseTrueVecOut[i] ?
16                        Max(vec[i], vecPermuted[i]):
17                        Min(vec[i], vecPermuted[i]))
18          //svzip1 - [F, F, T, T, F, F, T, T, …]
19          falseTrueVecOut = (i ∈ [0, N-1] → falseTrueVecOut[i/2])
20          //svsel/svadd/svsub - [3, 2, 1, 0, 3, 2, 1, 0, …]
21          vecIndexesPermOut = (i ∈ [0, N-1] → falseTrueVecOut[i] ?
22                        vecIndexesPermOut[i]-stepOut*2:
23                        vecIndexesPermOut[i]+stepOut*2)
24      end
25      return vec
```

We use the same principle in the *stair* stage, with one vector for the Boolean that shows the direction of the exchange and another to store the relative index for comparison. We sum this last vector with the index vector to get the permutation indices. If we study again to Fig. 2A, we observe that the algorithm starts by working on parts of half the size of the previous *symmetric* stage. Then, at each iteration, the parts are subdivided until they contain two elements. Besides, the width of the exchange is the same for all elements in an iteration and is then divided by two for the next iteration.

We provide a pseudo-code of our vectorized algorithm in Algorithm 2. To manage the Boolean vector: we use the *svuzp2* instruction that select odd elements from two inputs and concatenate them. In our case, we give a vector that contains a repeated pattern composed of *false* $x$ times, followed by *true* $x$ times ($x$ a power of two) to *svuzp2* to get a vector with repetitions of size $x/2$. Therefore, we pass the vector of Boolean generated during the *symmetric* stage to *svuzp2* to initialize the new Boolean vector (line 7). We divide the exchange step by two for all elements (line 23). The permutation (line 15) and exchange (line 17) are similar to what is performed in the *symmetric* stage.

---

**Algorithm 2:** SVE Bitonic sort for one vector, *stair* stage. The gray lines are copied from the *symmetric* stage (Algorithm 1)

```
Input: vec: a SVE vector to sort.
Output: vec: the vector sorted.
1  function sve_bitonic_sort_1v_stairs(vec)
2      N = get_hardware_size()
3      vecIndexes = (i ∈ [0, N-1] → i)
4      falseTrueVecOut = (i ∈ [0, N-1] → i is odd ? False: True)
5      for stepOut from 1 to N-1, doubling stepOut at each step do
6          //svuzp2
7          falseTrueVecIn = (i ∈ [0, N-1] → falseTrueVecOut[(i*2+1)%N])
8          //svdup - [stepOut/2, stepOut/2, ...]
9          vecIncrement = (i ∈ [0, N-1] → stepOut/2)
10         for stepIn from stepOut/2 to 1, dividing stepIn by 2 at each step do
11             //svadd/svneg - [stepOut/4, stepOut/4, ..., -stepOut/4, -stepOut/4]
12             premuteIndexes = (i ∈ [0, N-1] → vecIndexes[i] +
13                                 (falseTrueVecIn[i] ? -vecIncrement[i]: vecIncrement[i]))
14             //svtbl
15             vecPermuted = (i ∈ [0, N-1] → vec[premuteIndexes[i]])
16             //svsel/svmin/svmax
17             vec = (i ∈ [0, N-1] → falseTrueVecIn[i] ?
18                                 Max(vec[i], vecPermuted[i]):
19                                 Min(vec[i], vecPermuted[i]))
20             //svuzp2
21             falseTrueVecIn = (i ∈ [0, N-1] → falseTrueVecIn[(i*2+1)%N])
22             //svdiv
23             vecIncrement = (i ∈ [0, N-1] → vecIncrement[i] / 2);
24         end
25         falseTrueVecOut = (i ∈ [0, N-1] → falseTrueVecOut[i/2])
26     end
27     return vec
```

---

The complete function to sort a vector is a mix of the *symmetric* (*sve_bitonic_sort_1v_symmetric*) and *stair* (*sve_bitonic_sort_1v_stairs*) functions; each iteration of the *symmetric* stage is followed by the inner loop of the *stair* stage. The corresponding C++ source code of a fully vectorized implementation is given in Appendix A.1.

### Sorting more than one vectors

To sort more than one vector, we profit that the same patterns are repeated at different scales; to sort $V$ vectors, we re-use the function that sorts $V/2$ vectors and so on. We

provide an example to sort two vectors in Algorithm 3, where we start by sorting each vector individually using the *sve_bitonic_sort_1v* function. Then, we compare and exchange values between both vectors (line 9), and we finish by applying the same *stair* stage on each vector individually. Our real implementation uses an optimization that consists in a full inlining followed by a merge of the same operations done on different data. For instance, instead of two consecutive calls to *sve_bitonic_sort_1v* (lines 7 and 8), we inline the functions. But since they are similar but on different data, we merge them into one that works on both vectors at the same time. In our sorting implementation, we provide the functions to sort up to 16 vectors.

---

**Algorithm 3:** SIMD bitonic sort for two vectors of double floating-point values.

**Input**: vec1 and vec2: two double floating-point SVE vectors to sort.
**Output**: vec1 and vec2: the two vectors sorted with vec1 lower or equal than vec2.
1 **function** sve_bitonic_exchange_rev(vec1, vec2)
2     vec1_copy = (i ∈ [0, N-1] → vec1[N-1-i])
3     vec1 = (i ∈ [0, N-1] → Min(vec1[i], vec2[i]))
4     vec2 = (i ∈ [0, N-1] → Max(vec1_copy[i], vec2[i]))
5     return {vec1, vec2}
6 **function** sve_bitonic_sort_2v(vec1, vec2)
7     vec1 = sve_bitonic_sort_1v(vec1)
8     vec2 = sve_bitonic_sort_1v(vec2)
9     [vec1, vec2] = sve_bitonic_exchange_rev(vec1, vec2)
10     vec1 = sve_bitonic_sort_1v_stairs(vec1)
11     vec2 = sve_bitonic_sort_1v_stairs(vec2)

---

### Sorting small arrays

Once a partition contains less than 16 SIMD-vector elements, it can be sorted with our SVE-Bitonic functions. We select the appropriate SVE-Bitonic function (the one that matches the size of the array to sort) with a switch statement, in a function interface that we refer to as *sve_bitonic_sort_wrapper*. However, the partitions obtained from the QS do not necessarily have a size multiple of the vector's length. Therefore, we pad the last vector with an extra value, which is the greatest possible value for the target data type. During the execution of the sort, these last values will be compared but never exchanged and will remain at the end of the last vector.

### Optimization by comparing vectors' min/max values or whether vectors are already sorted

There are two main points where we can apply optimization in our implementation. The first one is to avoid exchanging values between vectors if their contents are already in the correct order, *i.e.* no values will be exchanged between the vectors because their values respect the ordering objective. For instance, in Algorithm 3, we can compare if the greatest value in vector *vec2* (SVE instruction *svmaxv*) is lower than or equal to the lowest value in vector *vec1* (SVE instruction *svminv*). If this is the case, the function can simply sort each vector individually. The same mechanism can be applied to any number of vectors, and it can be used at function entry or inside the loops to break when it is known that no more values will be exchanged. The second optimization can be applied when we want to sort a single vector by checking if it is already sorted. Similarly to the first optimization, this check can be done at function entry or in the loops, such as at lines 2 and 10, in Algorithm

2. We propose two implementations to test if a vector is sorted and provide the details in Appendix A.2.

## Partitioning with SVE

Our partitioning strategy is based on the AVX-512-partition. In this algorithm, we start by saving the extremities of the input array into two vectors that remain unchanged until the end of the algorithm. By doing so, we free the extremity of the array that can be overwritten. Then, in the core part of the algorithm, we load a vector and compare it to the pivot. The values lower than the pivot are stored on the left side of the array, and the values greater than the pivot are stored on the right side while moving the corresponding cursor indexes. Finally, when there is no more value to load, the two vectors that were loaded at the beginning are compared to the pivot and stored in the array accordingly.

When we implement this algorithm using SVE we obtain a Boolean vector $b$ when we compare a vector to partition with the pivot. We use $b$ to compact the vector and move the values lower or equal than the pivot on the left, and then we generate a secondary Boolean vector to store only as a sub-part of the vector. We manage the values greater than the pivot similarly by using the negate of $b$.

## Sorting key/value pairs

The sorting methods we have described are designed to sort arrays of numbers. However, some applications need to sort key/value pairs. More precisely, the sort is applied on the keys, and the values contain extra information such as pointers to arbitrary data structures, for example. We extend our SVE-Bitonic and SVE-Partition functions by making sure that the same permutations/moves apply to the keys and the values. In the sort kernels, we replace the minimum and maximum statements with a comparison operator that gives us a Boolean vector. We use this vector to transform both the vector of keys and the vector of values. For the partitioning kernel, we already use a comparison operator, therefore, we add extra code to apply the same transformations to the vector of values and the vector of keys.

In terms of high-level data structure, we support two approaches. In the first one, we store the keys and the values in two distinct arrays, which allow us to use contiguous load/store. In the second one, the key/value is stored by pair contiguously in a single array, such that loading/storing requires non-contiguous memory accesses.

## Parallel sorting

Our parallel implementation is based on the *QS-par* that we extend with several optimizations. In the *QS-par* parallelization strategy, it is possible to avoid having too many tasks or tasks on too small partitions by stopping creating tasks after a given recursive level. This approach allows to fix the number of tasks at the beginning, but could end in an unbalanced configuration (if the tasks have different workload) that is difficult to resolve on the fly. Therefore, in our implementation, we create a task for every partition larger than the L1 cache, as shown in Algorithm 4 line 26. However, we do not rely on the OpenMP task statement because it is impossible to control the data locality. Instead, we use one task list per thread (lines 2, 11 and 33). Each thread uses its list as a stack to store the

intervals of the recursive calls and also as a task list where each interval can be processed in a task. In a steady-state, each thread accesses only its list: after each partitioning, a thread puts the interval of the first sub-partition in the list and continues with the second sub-partition (line 35). When the partition is smaller than the L1 cache, the thread executes the sequential SVE-QS. We use a work-stealing strategy when a thread has an empty list such that the thread will try to pick a task in others' lists. The order of access to others' lists is done such that a thread accesses the lists from threads of closer ids to far ids, e.g. a thread of id $i$ will look at $i+1$, $i-1$, $i+2$, and so on. We refer to this optimized version as the *SVE-QS-par*.

---

**Algorithm 4:** Simplified algorithm of SVE-QS-par.

**Input**: array: data to sort.
```
1  function sve_par_sort(array, N)
2         buckets = init_buckets()
3         nb_threads_idle = 0
4         #pragma omp parallel
5              current_thread_idle = False
6              #pragma omp master
7                   core_par_sort(array, 0, N, buckets)
8              while nb_threads_idle ≠ nb_threads do
9                   //Try to get a task, from current thread's list
10                  //then neighbors' lists, etc.
11                  interval = steal_task(buckets)
12                  if interval is null then
13                       if current_thread_idle is False then
14                            current_thread_idle = True
15                            nb_threads_idle += 1
16                       end
17                  else
18                       if current_thread_idle is True then
19                            current_thread_idle = False
20                            nb_threads_idle -= 1
21                       end
22                       core_par_sort(array, interval.start, interval.end, buckets)
23                  end
24             end
25  function core_par_sort(array, start, end, buckets)
26         if (start-end) × size of element ≤ size of L1 then
27              //Sort sequentially
28              sve_bitonic_sort(array, start, end)
29         else
30              //Partition the array
31              p = sve_partition(array, start, end)
32              //Put first partition in the buckets
33              insert(buckets[current_thread_id()], p.second_partition)
34              //Directly work on first partition
35              core_par_sort(array, p.first_partition.start, p.first_partition.end, buckets)
36         end
```

---

# PERFORMANCE STUDY

## Configuration

We assess our method on an ARMv8.2 *A64FX - Fujitsu* with 48 cores at 1.8 GHz and 512-bit SVE, *i.e.* a vector can contain 16 integers and eight double floating-point values. The node has 32 GB HBM2 memory arranged in four core memory groups (CMGs) with 12 cores and 8GB each, 64KB private L1 cache, 8MB shared L2 cache per CMG. For the sequential executions, we pinned the process with *taskset -c 0*, and for the parallel executions, we use *OMP_PROC_BIND =TRUE*. We use the ARM compiler 20.3 (based on LLVM 9.0.1) with the aggressive optimization flag *-O3*. We compare our sequential implementations against

the GNU STL 20200312 from which we use the *std::sort* and *std::partition* functions. We also compare against SVE512-Bitonic, which is an implementation that we have obtained by performing a translation of our original AVX-512 into SVE. This implementation works only for 512-bit SVE, but this makes it possible to hard-code all the indices of the compare and exchange of the Bitonic algorithm. Moreover, SVE512-Bitonic does not use any loop, *i.e.* it can be seen as if we had fully unrolled the loops of the SVE-Bitonic.

We compare our parallel implementation against the Boost (https://www.boost.org/) 1.73.0 from which we use the *block_in direct_sort* function. The test file used for the following benchmark is available online (https://gitlab.inria.fr/bramas/arm-sve-sort) and includes the different sorts presented in this study, plus some additional strategies and tests[4]. Our QS uses a 5-values median pivot selection (whereas the STL sort function uses a 3-values median). The arrays to sort are populated with randomly generated values. Our implementation does not include the potential optimizations described in 'Optimization by comparing vectors' min/max values or whether vectors are already sorted' that can be applied when there is a chance that parts or totality of the input array are already sorted[5].

As it is possible to virtually change the size of the SIMD vectors at runtime, we evaluated if using different vector sizes (128 or 256) could increase the performance of our approach. It appears that the performance was always worse, and consequently we decided not to include these results in the current study.

## Performance to sort small arrays

We provide in Fig. 4 the execution times to sort arrays of 1 element to $16 \times VEC\_SIZE$ elements. This corresponds to at most 128 double floating-point values, or 256 integer values. We test all the sizes by step 1, such that we include sizes not multiple of the SIMD-vector's length. For more than 20 values, the SVE-Bitonic always delivers better performance than the STL. The speedup is significant and increases with the number of values to reach 5 for 256 integer values. The execution time per item increases every $VEC\_SIZE$ values because the cost of sorting is not tied to the number of values but to the number of SIMD-vectors to sort, as explained in 'Sorting small arrays'. For example, we have to sort two SIMD-vector of 16 values to process from 17 to 32 integers. Our method reaches a speedup of 3.6 to sort key/value pairs. To sort key/value pairs, we obtain similar performance if we sort pairs of integers stored contiguously or two arrays of integers, one for the keys and one for the values. Comparing our two SVE implementations, SVE-Bitonic appears more efficient than SVE512-bitonic, except for very small number of values. This means that considering a static vector size of 512 bits, with compare-exchange indices hard coded and no loops/branches, does not provide any benefit, and is even slower for more than 70 values. This means that, for our kernels, the CPU manages more easily loops with branches (SVE-bitonic) than a large amount of instructions without branches (SVE512-bitonic). Moreover, the hard-coded exchange indices are stored in memory and should be load to register, which appear to hurt the performance compared to building these indices using several instructions. Sorting double floating-points values or pairs of integers takes similar duration up to 64 values, then with more values it is faster to sort pairs of integers.

[4]It can be executed on any CPU using the Farm-SVE library (https://gitlab.inria.fr/bramas/farm-sve)

[5]This implementation is partially implemented in the branch *optim* of the code repository.

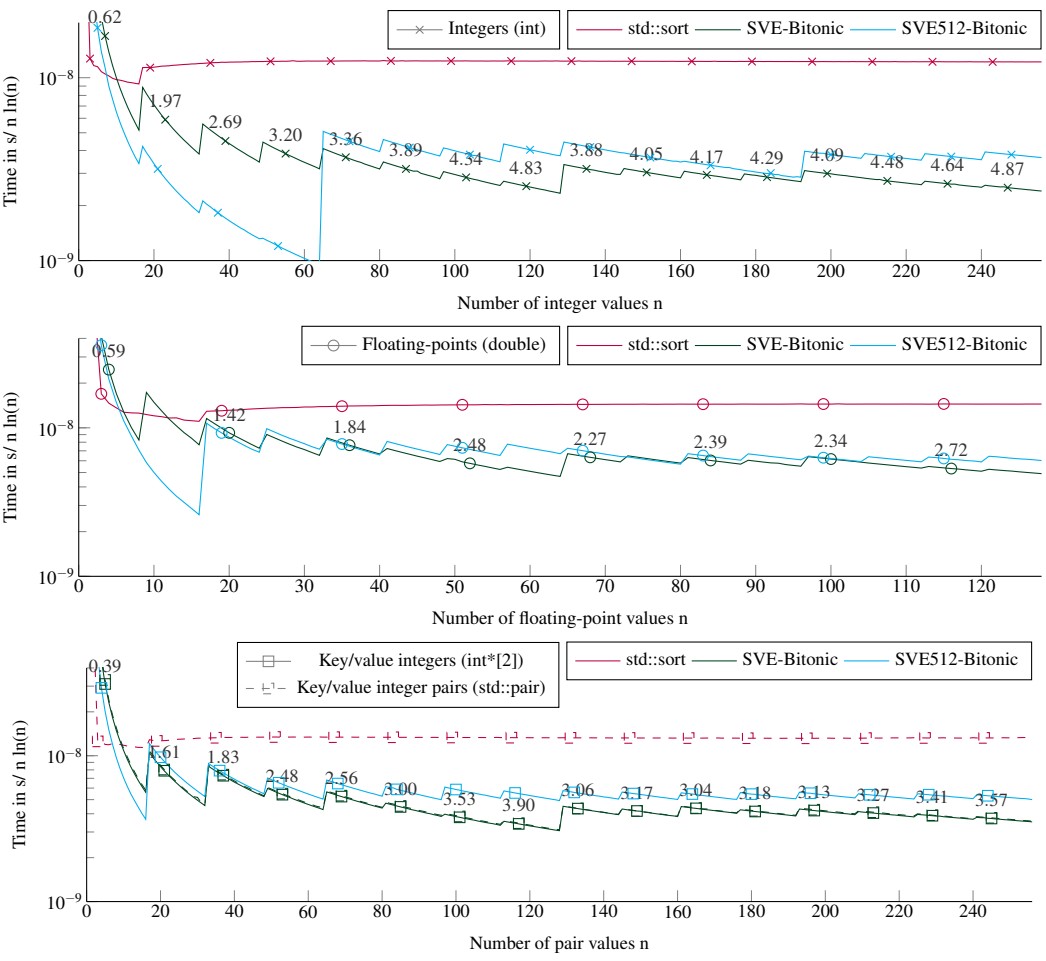

**Figure 4 Execution time to sort small arrays (in sequential).** Execution time divided by n ln(n) to sort from 1 to 16 × VEC_SIZE values. The execution time is obtained from the average of 2E3 sorts with different values for each size. The speedup of the SVE-Bitonic against the STL is shown above the SVE-Bitonic lines. Key/value integers as a std::pair are plot with dashed lines and as two distinct integer arrays (int*[2]) are plot with dense lines.

## Partitioning performance

Figure 5 shows the execution times to partition using our SVE-Partition or the STL's partition function. Our method provides again a speedup of an average factor of 4 for integers and key/values (with two arrays), and 3 for floating-point values. We see no difference if the data fit in the caches L1/L2 or not, neither in terms of performance nor in the difference between the STL and our implementation. However, there is a significant difference between partitioning two arrays of integers (one for the key and the other for the values) or one array of pairs of integers. The only difference between both implementations is that we work with distinct svint32_t vectors in the first one, and with svint32x2_t vector pairs in the second. But the difference is mainly in the memory accesses during the loads/stores. The partitioning of one array or two arrays of integers appears equivalent, and this can be unexpected because we need more instructions when managing the latter.

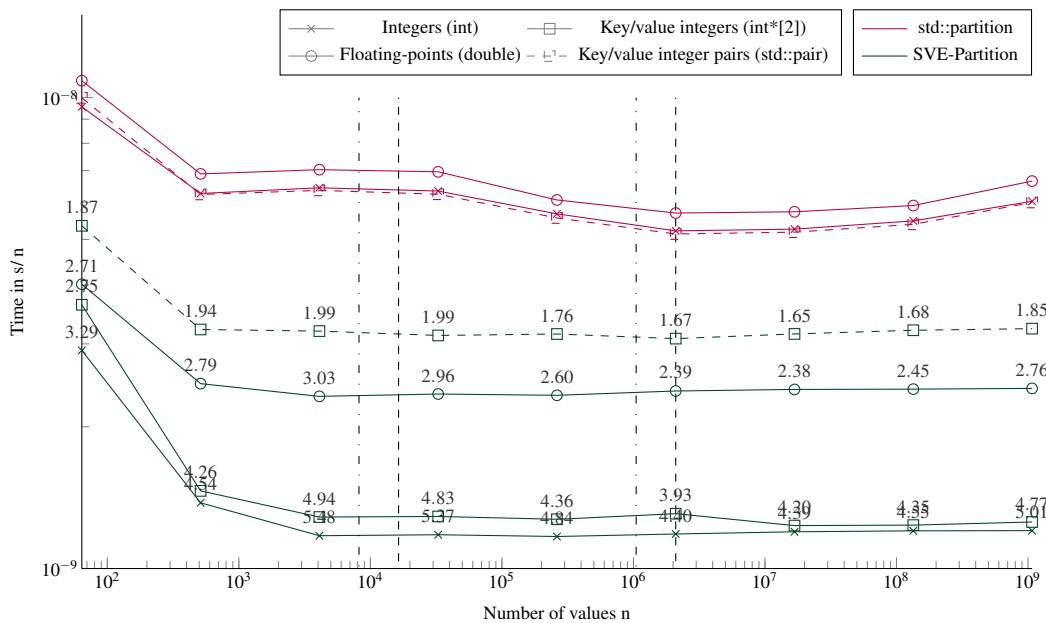

**Figure 5  Execution time to partition arrays (in sequential).** Execution time divided by n of elements to partition arrays filled with random values with sizes from 64 to 109 elements. The pivot is selected randomly. The execution time is obtained from the average of 20 executions with different values. The speedup of the SVE-partition against the STL is shown above the lines. The vertical lines represent the caches relatively to the processed data type (- for the integers and .-. for floating-points and the key/value integers). Key/value integers as a std::pair are plot with dashed lines and as two distinct integer arrays (int*[2]) are plot with dense lines.

Indeed, we have to apply the same transformations to the keys and the values, and we have twice memory accesses.

## Performance to sort large arrays

Figure 6 shows the execution times to sort arrays up to a size of $\approx 10^9$ items. Our SVE-QS is always faster in all configurations. The difference between SVE-QS and the STL sort is stable for size greater than $10^3$ values with a speedup of more than 4 to our benefit to sort integers. There is an effect when sorting 64 values (the left-wise points) as the execution time is not the same as the one observed when sorting less than 16 vectors (Fig. 4). The only difference is that here we call the main SVE-QS functions, which call the SVE-Bitonic functions after just one test on the size, whereas in the previous results we call the SVE-Bitonic functions directly. We observe that when sorting key/value pairs, there is again a benefit when using two distinct arrays of scalars compared with a single array of pairs. From the previous results, it is clear that this difference comes from the partitioning for which the difference also exists (Fig. 6), whereas the difference is negligible in the sorting of arrays smaller than 16 vectors (Fig. 4). However, as the size of the array increases, this difference vanishes, and it becomes even faster to sort Floating-point values than keys/values.

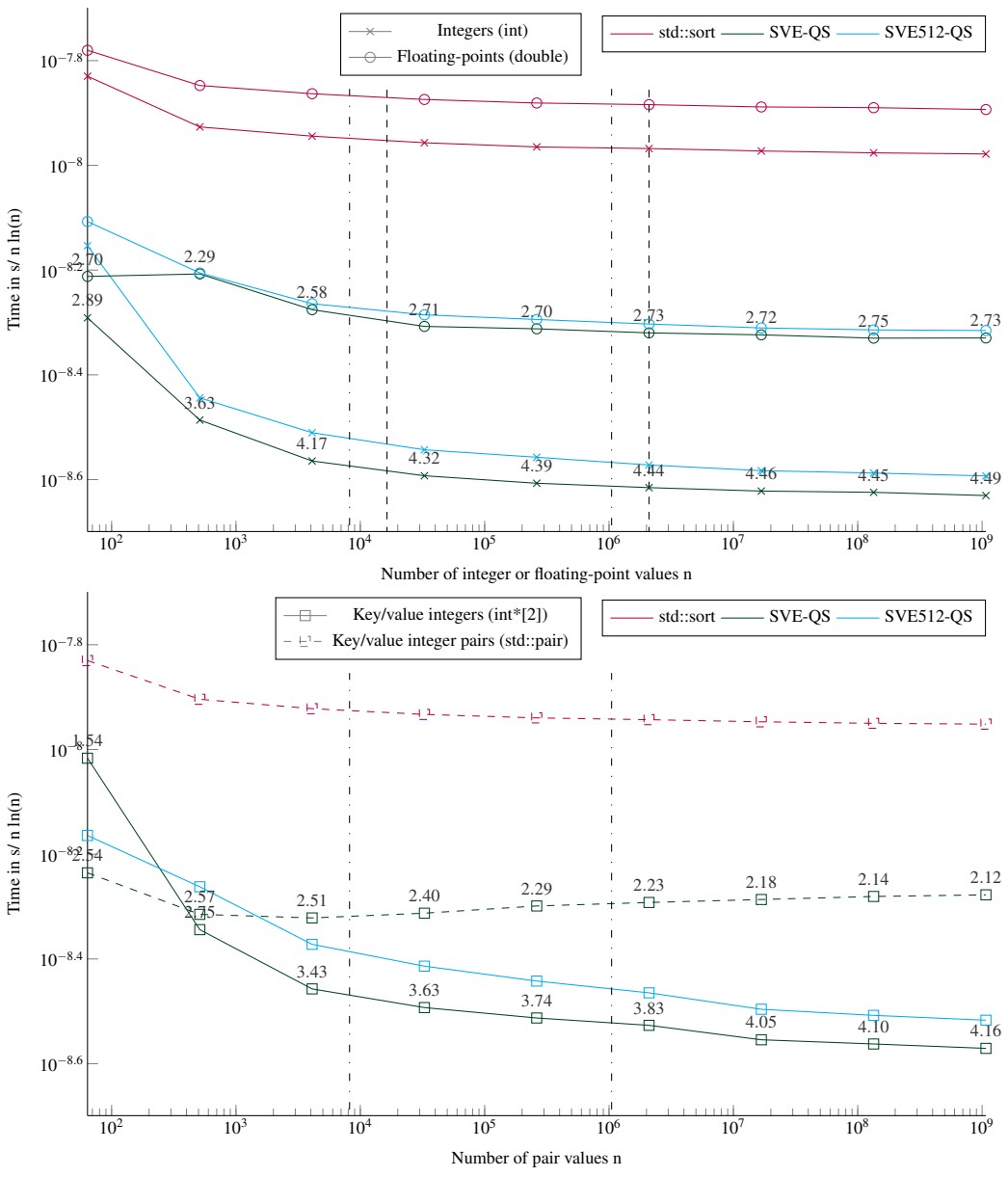

**Figure 6  Execution time divided by n ln(n) to sort arrays filled with random values with sizes from 64 to 109 elements.** The execution time is obtained from the average of 5 executions with different values. The speedup of the SVE-QS against the STL is shown above the SVE-QS lines. The vertical lines represent the caches relatively to the processed data type (- for the integers and .-. for floating-points and the integer pairs). Key/value integers as a std::pair are plot with dashed lines and as two distinct integer arrays (int*[2]) are plot with dense lines.

## Performance of the parallel version

Figure 7 shows the performance for different number of threads of a parallel sort implementation from the boost library (block_indirect_sort) against our task-based implementation (SVE-QS-par). Our approach is faster in all configurations, but the results

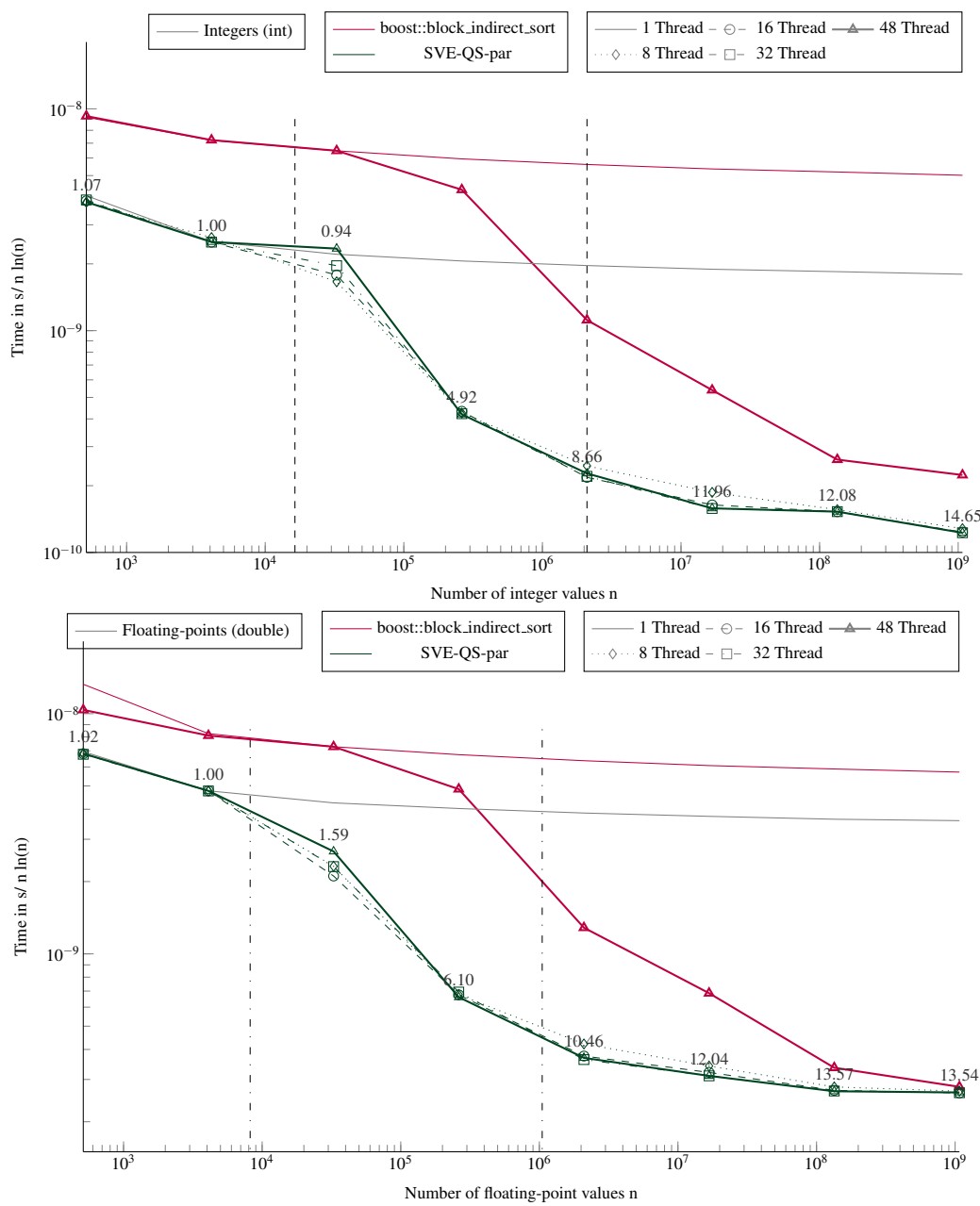

**Figure 7** **Execution time to sort large arrays (in parallel).** Execution time divided by n ln(n) to sort in parallel arrays filled with random values with sizes from 512 to ~109 elements. The execution time is obtained from the average of 5 executions with different values. The speedup of the parallel SVE-QS-par against the sequential execution is shown above the lines for 16 and 48 threads. The vertical lines represent the caches relatively to the processed data type (- for the integers and .-. for the floating-points).

show the benefit of using a merge strategy to sort large arrays, as in block_indirect_sort. Indeed, as the number of threads increases, the SVE-QS-par becomes faster but reaches a limit. Using more than 8 threads (16, 32 or 48 threads) does not provide any benefit, and the four curves for 8, 16, 32, and 48 threads, overlap or are very close for both data

types. Moreover, while 1 thread executions show that our SVE-QS-par is faster for both data types, for large arrays the block_indirect_sort provides very close performance. This illustrates the limit of the divide-and-conquer parallelization strategy to process large arrays since the first steps, *i.e.* the partitioning steps, are not or poorly parallel. Moreover, using more than 12 threads (the number of threads per CMG), while working in place on the original array implies memory transfers across the memory nodes, which impacts the scalability. Whereas the block_indirect_sort, which uses a merge kernel but at the cost of using additional data buffers, has a more stable scalability. Nevertheless, our approach is trivial to implement and delivers high performance when using few threads, which is valuable when an application uses multiple processes per node.

### Comparison with the AVX-512 implementation

The results obtained in our previous study on Intel Xeon Platinum 8170 Skylake CPU at 2.10 GHz (*Bramas, 2017*) shows that our AVX-512-QS was sorting at a speed of $\approx 10^{-9}$ second per element (obtained by $T/N \cdot ln(N)$). This was almost 10 times faster than the STL ($10^{-8}$ second per element). The speedup obtained with SVE in the current study is lower and does not come from our new implementation, which is generic regarding the vector size, because the SVE512-QS is not faster, either. The difference does not come either from the memory accesses, because it is significant for small arrays (that fit in the L1 cache), or the number of vectorial registers, which is 32 for both hardware. Profiling the code reveals that the cycles per instruction is around 1.7 for both SVE512-QS and SVE-QS in sequential, which is not ideal. The L2 cache miss rate is lower than 10%, which indicates that the memory access pattern is adequate. The memory bandwidth is given in Table 1. We observe that the peak of HBM2 (256GB/s) is not reached. Additionally, the table indicates that to sort 4GB of data, our approach will read/write 252GB of data, but it was already the case for AVX-512-QS. From the hardware specification of the A64FX (*Fujitsu, 0000*), we can observe that most SIMD SVE instructions have a latency between 4 and 9 cycles. Therefore, we conclude that the difference between our AVX-512 and SVE versions comes from the cost of the SIMD instructions and the pipelining of these because the memory access appears fine, and the difference is already significant for small arrays.

## CONCLUSIONS

In this paper, we described new implementations of the Bitonic sorting network and the partition algorithm that have been designed for the SVE instruction set. These two algorithms are used in our Quicksort variant, which makes it possible to have a fully vectorized implementation. Our approach shows superior performance on ARMv8.2 (A64FX) in all configurations against the GNU $C$++ STL. It provides a speedup up of five when sorting small arrays (less than 16 SIMD-vectors), and a speedup above four for large arrays. We also demonstrate that our algorithm is less efficient when we fully unroll the loops and use hard-coded exchange indices in the Bitonic stage (by considering that the vector if of size 512bits). This strategy was efficient when implemented with AVX512 and executed on Intel Skylake. Our parallel implementation is efficient, but it could be improved when working on large arrays by using a merge on sorted partitions instead of

**Table 1** Amount of memory accessed and corresponding bandwidth for SVE-QS and SVE512-QS to sort arrays of integers of size $N$. The accesses are measured by capturing the calls to SIMD loads/stores.

| $N$ | Size (GB) | SVE-QS | | SVE512-QS | |
|---|---|---|---|---|---|
| | | Memory read/write (GB) | Bandwith (GB/s) | Memory read/write (GB) | Bandwith (GB/s) |
| $2^6$ | 2.56E−07 | 5.12E−07 | 3.76E−01 | 2.75E−06 | 1.471 |
| $2^9$ | 2.05E−06 | 1.38E−05 | 1.325 | 3.88E−05 | 3.370 |
| $2^{12}$ | 1.64E−05 | 2.26E−04 | 2.427 | 3.96E−04 | 3.768 |
| $2^{15}$ | 1.31E−04 | 2.67E−03 | 3.064 | 4.19E−03 | 4.289 |
| $2^{18}$ | 1.05E−03 | 2.95E−02 | 3.651 | 4.05E−02 | 4.467 |
| $2^{21}$ | 8.39E−03 | 3.10E−01 | 4.192 | 3.98E−01 | 4.870 |
| $2^{24}$ | 6.71E−02 | 2.947 | 4.420 | 3.644 | 4.995 |
| $2^{27}$ | 5.37E−01 | 27.726 | 4.645 | 33.051 | 5.087 |
| $2^{30}$ | 4.294 | 252.233 | 4.822 | 299.740 | 5.257 |

a recursive parallel strategy (at a cost of using external memory buffers). In addition, we would like to compare the performance obtained with different compilers because there are many ways to transform and optimize a C++ code with intrinsics into a binary.

Besides, these results is a good example to foster the community to revisit common problems that have kernels for x86 vectorial extensions but not for SVE yet. Indeed, as the ARM-based architecture will become available on more HPC platforms, having high-performance libraries of all domains will become critical. Moreover, some algorithms that were not competitive when implemented with x86 ISA may be easier to vectorize with SVE, thanks to the novelties it provides, and achieve high-performance. Finally, the source code of our implementation is publicly available and ready to be used and compared against.

# APPENDIX

## A.1 Source code of sorting one vector of integers

In Code 1, we provide the implementation of sorting one vector using Bitonic sorting network and SVE.

```
1  inline void Sort1Vec(svint32_t& vecToSort){
2      const int N = svcntw(); // Number of values in a vector
3      const svint32_t vecIndexes = svindex_s32(0, 1); // [O, 1, ..., N−1]
4      svbool_t falseTrueVecOut = svzip1_b32(svpfalse_b(),svptrue_b32()); // [F, T, F, ..., T]
5      svint32_t vecIndexesPermOut = svsel_s32(falseTrueVecOut, svdup_s32(−1), svdup_s32(1));
6      for(long int stepOut = 1 ; stepOut < N ; stepOut *= 2){
7          {
8              const svint32_t premuteIndexes = svadd_s32_z(svptrue_b32(), vecIndexes, vecIndexesPermOut);
9              const svint32_t vecToSortPermuted = svtbl_s32(vecToSort, svreinterpret_u32_s32(premuteIndexes));
10             vecToSort = svsel_s32(falseTrueVecOut,
11                         svmax_s32_z(svptrue_b32(), vecToSort, vecToSortPermuted),
12                         svmin_s32_z(svptrue_b32(), vecToSort, vecToSortPermuted));
13         }
14         svbool_t falseTrueVecIn = svuzp2_b32(falseTrueVecOut,falseTrueVecOut); // [F, F, ..., T, T]
15         svint32_t vecIncrement = svdup_s32(stepOut/2);
16         for(long int stepIn = stepOut/2 ; stepIn >= 1 ; stepIn/=2){
17             const svint32_t premuteIndexes = svadd_s32_z(svptrue_b32(), vecIndexes,
18                         svsel_s32(fftt, svsel_s32(falseTrueVecIn, svneg_s32_z(falseTrueVecIn,
19                         vecIncrement), vecIncrement), vecIncrement));
20             const svint32_t vecToSortPermuted = svtbl_s32(vecToSort, svreinterpret_u32_s32(premuteIndexes));
21             vecToSort = svsel_s32(falseTrueVecIn,
22                         svmax_s32_z(svptrue_b32(), vecToSort, vecToSortPermuted),
23                         svmin_s32_z(svptrue_b32(), vecToSort, vecToSortPermuted));
24             falseTrueVecIn = svuzp2_b32(falseTrueVecIn,falseTrueVecIn);
25             vecIncrement = svdiv_n_s32_z(svptrue_b32(), vecIncrement, 2);
26         }
27         falseTrueVecOut = svzip1_b32(falseTrueVecOut,falseTrueVecOut);
28         vecIndexesPermOut = svsel_s32(falseTrueVecOut,
```

```
29                              svsub_n_s32_z(svptrue_b32(), vecIndexesPermOut, stepOut*2),
30                              svadd_n_s32_z(svptrue_b32(), vecIndexesPermOut, stepOut*2));
31      }
32  }
33
```

Code 1: Implementation of sorting a single SVE vector of integers.

### A.2 Source code to check if a vector of integers is sorted

In Code 2, we provide two implementations to test if a vector is already sorted. These functions can be used if there is a chance that parts or totality of the input vector are already sorted.

```
1   inline bool IsSorted(const svint32_t& input){
2       // Methode 1: 1 vec op, 1 comp, 2 bool vec op, 2 bool vec count
3       svint32_t revinput = svrev_s32(input);
4       svbool_t mask = svcmpgt_s32(svptrue_b32(), input, revinput);
5       svbool_t v1100 = svbrkb_b_z(svptrue_b32(), mask);
6       return svcntp_b32(svptrue_b32(),svnot_b_z(svptrue_b32(),v1100)) == svcntp_b32(svptrue_b32(),mask);
7   }
8
9   inline bool IsSorted(const svint32_t& input){
10      // Methode 2: 1 vec op, 1 comp, 2 bool vec op, 1 bool vec count
11      svbool_t FTTT = svnot_b_z(svptrue_b32(), svwhilelt_b32_s32(0, 1));
12      svint32_t compactinput = svcompact_s32(FTTT, input);
13      const size_t vecSizeM1 = (svcntb()/sizeof(int))-1;
14      svbool_t TTTF = svwhilelt_b32_s32(0, vecSizeM1);
15      svbool_t mask = svcmple_s32(svptrue_b32(), input, compactinput);
16      return svcntp_b32(TTTF,mask) == vecSizeM1;
17  }
18
```

Code 2: Possible implementations to test if a vector of integers is already sorted.

## ACKNOWLEDGEMENTS

This work used the Isambard 2 UK National Tier-2 HPC Service (http://gw4.ac.uk/isambard/) operated by GW4 and the UK Met Office, which is an EPSRC project (EP/T022078/1). In addition, this work used the Farm-SVE library (*Bramas, 2020*).

### Funding

The author received no funding for this work

### Competing Interests

The author declares there are no competing interests.

### Author Contributions

- Bérenger Bramas conceived and designed the experiments, performed the experiments, analyzed the data, performed the computation work, prepared figures and/or tables, authored or reviewed drafts of the paper, and approved the final draft.

### Data Availability

The source code and the scripts to reproduce the results are publicly available at GitLab: https://gitlab.inria.fr/bramas/arm-sve-sort.

## Supplemental Information

Supplemental information for this article can be found online at http://dx.doi.org/10.7717/peerj-cs.769#supplemental-information.

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
