# Peer review of "A fast vectorized sorting implementation based on the ARM scalable vector extension (SVE)"

_PeerJ Computer Science, doi:10.7717/peerj-cs.769_

## Round 0.1 · original submission · Major Revisions

Based on reviewers’ comments, you may resubmit the revised manuscript for further consideration. Please consider the reviewers’ comments carefully and submit a list of responses to the comments along with the revised manuscript.

Reviewer 1 ·

Basic reporting

This paper proposes a fast vectorized sorting implementation based on the ARM scalable vector extension (SVE). It ports the author's previous work about the AVX-SORT algorithm to SVE, defines a new Bitonic-sort variant using SVE, and implements an efficient Quicksort variant using OpenMP tasks.

Strengths:

The description is clear. The work seems technically sound and leads to impressive experimental results with raw data shared.

Weaknesses:

(1) It is good the author declares this work is to port the previous work in AVX to SVE and cites the paper "A Novel Hybrid Quicksort Algorithm Vectorized using AVX-512 on Intel Skylake". There is no cover letter so that the reviewer is not certain whether it is allowable to directly reuse figures 1, 2 and 3. Besides, it is not clear the difference between this paper and the reference paper.

(2) This paper is mainly about SVE, so it is better to not only mention AVX in the related work.

(3) The performance figures, especially figure 4 and figure 7, are very hard to read; e.g., there are two markers that are too similar in figure 4, and lines with the same color but different width in figure 7.

(4) As declared in the paper, there are three contributions. It would be good if the author could detail the last one, "Implement an efficient Quicksort variant using OpenMP tasks", because there are only very brief descriptions about this in the paper.

(5) What's the difference between SVE-Bitonic and SVE512-Bitonic? Is SVE512-Bitonic a version of SVE-Bitonic with fixed vector length? If so, the performance is a little confusing as the performance of SVE512-Bitonic will not be worse than SVE-Bitonic, right?

(6) In figure 7, there are several lines of different numbers of threads are overlapped, which means the overheads and benefits of different numbers of threads on the same problem are the same. The reviewer was wondering why this happens.

(7) It would be a plus if the author could define how good the improvement is by using SVE in this paper, e.g., comparing it to the theoretical one instead of only to AVX.

Experimental design

no comment

Validity of the findings

no comment

Additional comments

no comment

Reviewer 2 ·

Basic reporting

1. This paper need to add more information about Scalable Vector Extension in introduction and related part. Especially in section 2.2 and 2.3. It feels like the author is writing a paper about AVX instead of SVE. In section 2.2.1 why use avx512 as an example instead of SVE? SVE related work and publications are not mentioned in section 2.3.

2. Figure 4 is very confusing and hard to read. Choose different lines and markers will be better.

Experimental design

1. In line 365 the author claims that "This means that considering a static vector size of 512 bits, with compare-exchange indices hard coded and no loops/branches, does not provide any benefit, and is even slower for more than 70 values."

The author needs to provide more information and details to support this assumption, because normally extra branching instruction may need extra cycles. It will be helpful if the author can provide low level explanation, such as assemble code snippets. Also some performance tools

2. The vector length of A64FX is configurable, I would like to see the results of the same experiments by setting vector length to 256 bits, then we can evaluate performance benefits with different vector length.

Validity of the findings

"We also demonstrate that an implementation designed for a fixed size of vectors is less
efficient", I would like to see more evidence to support this conclusion.

---

## Round 0.2 · Minor Revisions

Nearly all concerns have been addressed. Please address the last few comments of the reviewer.

Reviewer 1 ·

Basic reporting

Thanks for the authors' detailed response, and there is one remaining question. In the description of Figure 7, it's said "using more than 8 threads does not provide any benefit", which is not true when the number of values is large, as shown in the figure.

Minor:

(1) Figure 7 is split so that maybe markers could be used to differentiate lines instead of line width.

(2) there is only one subsubsection (2.2.1), and maybe it's better to remove it.

Experimental design

no comment

Validity of the findings

no comment

---

## Round 0.3 · accepted · Accept

Congratulations, the revised manuscript version is accepted for publication.